# Endovascular Arteriovenous Fistula Creation—Review of Current Experience

**DOI:** 10.3390/diagnostics12102447

**Published:** 2022-10-10

**Authors:** Alexandros Mallios, Jan Malik, William C. Jennings

**Affiliations:** 1Department of Vascular Surgery, Groupe Hospitalier Paris Saint Joseph, 75014 Paris, France; 2Department Internal Medicine and General University Hospital, First Medical Faculty, Charles University, U Nemocnice 1, 128 08 Prague, Czech Republic; 3Department of Surgery School of Community Medicine, Tulsa, OK 74135, USA

**Keywords:** EndoAVF, percutaneous AVF, Ellipsys, WavelinQ

## Abstract

Functioning vascular access is an essential element for life-saving hemodialysis therapy. A surgically-created arteriovenous fistula has been considered the best option for many years. Recently, two manufacturers developed systems for percutaneous/endovascular creation of an arteriovenous fistula (WavelinQ and Ellipsys). We provide a review of the available experience with these systems and discuss advantages and disadvantages.

## 1. Introduction

There are few examples of disruptive new technology developed in the field of vascular access for hemodialysis therapy, and possibly none that compare with the development of devices for the creation of a hemodialysis arteriovenous fistula (AVF) using percutaneous techniques. The WavelinQ^TM^ system (former EverlinQ by TVA Medical, Becton Dickinson, Franklin Lakes, NJ, USA) and the Ellipsys^®^ system (Avenu Medical, San Juan Capistrano Calif., recently acquired by Medtronic, Minneapolis, MN, USA), are distinctly different device systems developed separately with the same goal of establishing safe and reliable autogenous vascular access but with very different approaches, techniques and mechanisms. This article aims to provide a thorough technical review of both systems as well as a review of recently published results in the medical literature [1,2,3,4,5,6,7].

## 2. Techniques—WavelinQ^TM^

Inspired by traumatic AVFs that might develop accidentally due to trauma, vascular interventions or cannulations, the WavelinQ device creates an arteriovenous communication between the ulnar artery and ulnar vein or radial artery and radial vein a few centimetres distal to the perforating vein of the elbow (PVE). The device consists of two 4Fr catheters (arterial and venous) placed with ultrasound guidance and aligned under fluoroscopy with the help of rare earth magnets and markers that indicate the appropriate orientation (Figure 1 and Figure 2) [8,9,10,11,12].

Considered an advantage of this device, the artery and especially the vein can be accessed at variable locations based on the planned site of the AVF. Instructions for use (IFU) in the United States require arterial catheter insertion to be placed in the brachial artery, while in Europe and other locations allow arterial access through the radial or ulnar arteries. Ultrasound guidance is always used for the puncture, and fluoroscopy is required for catheter positioning. Many European practitioners prefer arterial access to the radial or ulnar artery at the level of the wrist to avoid the risk of brachial artery complications. The 5Fr slender sheath used for both catheters provides safe and efficient device deployment. The venous catheter insertion site options are more versatile as deep and/or superficial veins can be accessed based on anatomy and operators’ preference. Puncturing the distal ulnar or radial veins can be challenging because of their size and potential for spasm, but it is advantageous because guidewire advancement and catheter positioning are straightforward. Accessing a brachial vein or even a superficial vein such as the cephalic or the basilic is also possible and less cumbersome in terms of size. However, these more proximal venous access sites may entail the challenge of navigating the wire against the valves when advancing the catheter distally towards the desired AVF location.

When the catheters are judged to be appropriately positioned under fluoroscopy, activation of the generator will deliver a radiofrequency current, creating communication between the artery and the vein. Most of the time, this will be located a few centimetres below the PVE, with a variable amount of flow directed to the deep venous network. To minimise this deep venous AVF flow, placing a coil one of the two brachial veins adjacent to the AVF is specified in the IFU to prioritise flow through the PVE towards the superficial veins that are targeted for maturation and eventual cannulation for dialysis.

A proximal ulnar artery to vein AVF may create higher initial flows but is deeper and adjacent to the median nerve, resulting in a somewhat higher risk if a complication with the device occurs or if a surgical reintervention is required. A radial-to-radial configuration offers a safer profile, but the vessels may be smaller and more prone to spasm. Careful preoperative ultrasound evaluation is performed by the operator before and after regional bloc anaesthesia, allowing selection of the best possible location for the arterio-venous connection and the planned cannulation zone.

## 3. Techniques—Ellipsys^®^

Percutaneous AVF creation with Ellipsys system share some common characteristics with WavelinQ but also has some critical differences. Like the WavelinQ device, it relies on a patent, good quality PVE for efficient flow distribution to the superficial veins for establishing safe and reliable dialysis. Additional venous outflow branches are also common with Ellipsys, but the need for deep vein ligation or coiling is far less important than with WavelinQ [13,14,15,16,17,18,19]. AVF creation with the Ellipsys system utilises only ultrasound imaging without the need for fluoroscopy and radiation exposure for the patient and providers.

The Ellipsys device creates an anastomosis between the proximal radial artery (PRA) and the PVE at their junction via tissue fusion (pressure and thermal energy), resulting in a secure connection of the arterial and venous walls (Figure 3). It establishes a “Gracz” or “Jennings” AVF with a location exactly at the level of the perforator and almost exclusively with radial artery inflow. This may explain the very uncommon need for deep vein ligation or coiling. The Ellipsys device has excellent characteristics in terms of safety and freedom from complications.

A descriptive video of the procedure can be accessed online [19]. Briefly, under ultrasound guidance, single venous access (most commonly the median cephalic vein) is accessed with a 21G echogenic needle approximately 3–4 cm above the intended location of the anastomosis. This allows enough length for the balloon dilatation of the anastomosis performed immediately after the creation of the anastomosis and, at the same time, allows enough leverage to manoeuvre the needle through the PEV to meet the artery. With careful and slow movements under ultrasound guidance, the needle is advanced down the PEV until it is adjacent to the PRA. We prefer the transverse view or combination of transverse and longitudinal ultrasound visualisations. Always being aware of the location of the needle tip is the key element for success. Crossing from the vein to the artery is generally easier with transverse ultrasound visualisation, as the PVE and the PRA are not strictly parallel structures but cross pathways at the typical anastomotic site. It is easier to maintain the position of the needle tip in the centre of the PVE and then cross from the vein to the artery while slowly advancing the needle and the ultrasound probe simultaneously. A longitudinal visualisation of the needle at the crossing point will often lead the operator into a side wall arterial cannulation, creating a hematoma and inducing spasm that will obstruct further visualisation and make the completion of the procedure much more difficult. We found this to be the most common hurdle for inexperienced operators.

Once the puncture of the PRA through the PVE is achieved, a 0.021-inch wire is advanced into the distal radial artery, and the position is confirmed by ultrasound. A 6Fr Slender sheath is advanced over the wire and exchanged for a 0.014-inch wire. The Ellipsys catheter is then advanced and positioned at the junction of the vein with the artery. Under ultrasound guidance, the vessel walls are captured by the two edges of the Ellipsys device, and a generator delivers three rapid circles of heat and pressure, creating the anastomosis with tissue fusion. An immediate angioplasty of the anastomosis with a 5 × 20 balloon over the wire after removal of the Ellipsys catheter is now standard of care and was found to be crucial in avoiding early thromboses and improving rates and time of maturation [15,19].

Another important element for success has been prompt follow-up and intervention when necessary. Figure 4 shows a general algorithm to avoid maturation failure. A brachial artery flow volume >600 mL/min is recommended prior to initial cannulation. If the operative flow rate is less than 300 mL/min or clinical concerns are noted after completion of the procedure, such as a marginal thrill with palpation, the patient should be evaluated in one week for ultrasound imaging, flow rate measurement and a possible fistulogram. If not clearly maturing, angioplasty of the anastomosis and/or pre-anastomotic radial artery should be considered (Figure 5). Individuals with flow rates >600 mL/min and clear clinical success may be seen for follow-up at four weeks, with anticipated cannulation soon thereafter. Clinical targets must also include adequate cannulation zone length, diameter and depth.

## 4. Results of Clinical Trials

A multicenter single-arm prospective trial (NEAT) conducted in Canada, Australia and New Zealand is the most cited study of the WavelinQ system commercialised at the time as EverlinQ by TVA Medical [21]. This initial report included 80 patients (57% pre-dialysis) treated with the first 6Fr generation catheters. Technical success was 98%. The primary endpoint, defined as clinical (successful two needle dialysis) or physiological (brachial artery flow of greater than or equal to 500 cc/min and vein diameter greater than or equal to 4 mm), was achieved in 87% of patients. Twelve-month primary patency was 69%, and cumulative patency was 84%. Functional usability was 64% in patients on dialysis. Procedure-related adverse events were 8%, with the majority being hematomas with two pseudoaneurysms reported, probably related to brachial access and the larger profile of the device (6Fr). Secondary procedures (0.46/patient–year) were reported as transpositions, coil embolisations, ligations, thrombin injections, angioplasties, thrombectomies, surgical artery repairs and new access creations.

Berland et al. recently published aggregated results from three different studies regarding the current 4Fr generation of the WavelinQ system [22]. A total of 120 patients were included, with a technical success rate of 96.7%. Primary, assisted-primary and secondary 6-month patency rates were 71.9% ± 4.5%, 80.7% ± 4.1% and 87.8% ± 3.3%, respectively. Time to maturation averaged 41 ± 17 days and time to successful cannulation averaged 68 ± 51 days. Device and procedure-related serious adverse events were 2.5% and 5.8%, respectively. Arterial or venous access complications were not reported. Reinterventions were performed in 23 patients (19.2%), split between those performed for EndoAVF maturation (13/120 [10.8%]) and maintenance (11/120 [9.2%]).

Inston et al. compared the endovascular WavelinQ AVF (W group, *n* = 30) with surgically created distal radio-cephalic AVFs (group RC, *n* = 40) [23]. Technical success was 96.7% for the WavelinQ W group and 92.6% for the surgical RC group. Primary patency at 6 and 12 months was greater in group W (65.5% 6 mo and 56.5% 12 mo) vs. RC (53.4% 6 mo and 44% 12 mo) (*p* = 0.69 and 0.63). Mean primary patency was significantly lower for RC (235 ± 210 days) vs. W (362 ± 240 days) (*p* < 0.05). Secondary patency for group W was 75.8% and 69.5% at 6 and 12 months, respectively. Secondary patency for RC was lower at 66.7% and 57.6% at 6 and 12 months, respectively. They conclude that at their centre, WavelinQ appears superior to distal RC AVFs. Particularly, in patients with lower quality vessels at the level of the wrist, WavelinQ may be considered the first AVF choice.

We have published our extensive experience with the Ellipsys Vascular access system reporting our results and recommendations for maintenance procedures [20]. Technical success was achieved in 232 individuals (99%), and the average duration of the procedure was 15 min (7–35 min). The average follow-up was 252 days (range, 83–696 days). The 1-year primary, assisted-primary and secondary patency rates were 54%, 85% and 96%, respectively, comparing well with the largest series of surgically created PRA AVFs [24]. The average pAVF flow was 923 mL/min (range, 425–1440 mL/min). There were no significant adverse events related to the procedure. Only three patients (1%) required a later conversion of the pAVF anastomosis to a surgical fistula. Twenty-four (10%) patients required superficialisation of deep outflow veins because of difficult cannulation. The average maturation time was 4 weeks (range, 1–12 weeks). Fourteen patients (6%) had early cannulation of the pAVF (<2 weeks after creation).

Hull et al. reported results of maturation procedures from the Ellipsys post-market registry [25]. At 4 weeks, 67% (40 of 60) fistulae had undergone maturation procedures: 62% (37 of 60) had balloon dilation, 32% (19 of 60) had brachial vein embolisation, and 30% (18 of 60) had median cubital vein banding, increasing targeted outflow flow volume from 182 ± 123 mL/min to 572 ± 225 mL/min (*p* < 0.0005). Transposition or elevation procedures were required in 33% of patients (20 of 60), reducing the mean targeted cannulation zone depth from 10.9 to 3.7 mm (*p* < 0.0001). Two needle cannulation for three consecutive sessions was achieved in 87% (47 of 54); 10% of patients (6 of 60) were not on dialysis; 6.8% of patients (4 of 60) died; 5% of fistulas (3 of 60) were abandoned for arm swelling, steal syndrome, or thrombosis. Time to two needle cannulation, fistula success, and tunnelled catheter removal were 65.6 ± 45.7 days, 79.1 ± 50.9 days and 113.4 ± 62 days, respectively. Patients achieving successful cannulation had brachial artery flow of 944 ± 284 mL/min; and target vein flow, diameter and depth of 674 ± 292 mL/min, 6.1 ± 0.8 mm and 3.6 ± 1.3 mm, respectively.

Shahverdyan et al. compared pAVF created with Ellipsys vs. WavelinQ and, in a second report, compared percutaneous AVFs with Gracz surgical AVFs [26,27]. In the first study [26], they compared 65 patients with Ellipsys vs. 35 patients with WavelinQ pAVF. Technical success was similar: 100% vs. 97%, but median procedure times were shorter for Ellipsys (14 vs. 63 min, respectively (*p* < 0.001)). Maturation at 4 weeks for Ellipsys was 68.3% vs. 54.3% for WavelinQ, and median times to cannulation were 60 (1–164) vs. 90 (1–180) days, respectively. Successful pAVF dialysis was established in 31 of 39 patients (79.5%) for Ellipsys versus 14 of 24 patients (58%) for WavelinQ (*p* = 0.071). Access-related adverse events were more common for WavelinQ (3 vs. 1). Reinterventions were required in roughly 27% of both groups, but the number of interventions per patient-years was higher for Ellipsys (0.96 vs. 0.46, respectively). AVF failure was seen in 15.4% created by the Ellipsys system versus 37.1% in the WavelinQ group (*p* = 0.0137). Secondary patency at 12 months was significantly higher among patients who had an Ellipsys procedure (82%) than among those who underwent the WavelinQ procedure (60%).

In the second study [27], Shahverdyan and colleagues compared percutaneous AVFs with surgical Gracz-type AVFs. Surgical and percutaneous procedures had similar results with regard to maturation time and flow volume. The incidence of primary patency failure at 12 months was lower for surgical AVFs (47% vs. 64%, *p* = 0.1), but secondary patency failure was not different between groups (20% vs. 12%, *p* = 0.3). Surgical AVFs that used the proximal radial artery had similar primary patency (65% vs. 64%, *p* = 0.8) but higher secondary patency failure rates than percutaneous AVFs at 12 months (34% vs. 12%, *p* = 0.04). They noted that for the surgical group, 58% of patients had brachial artery inflow, while the proximal radial artery provided inflow for all percutaneous AVFs.

We also compared our percutaneous fistulas (pAVF) with surgical AVF (sAVF) in a single-centre observational study [28]. Two equivalent groups of 107 patients each were matched well in terms of demographic data and comorbidities except for the number that were already on dialysis (61% for pAVF vs. 47% for sAVF; *p* < 0.05). Overall, *p*-AVFs showed superior maturation rates at 6 weeks (65% vs. 50%; *p* = 0.01). The primary patency rates were greater for the s-AVFs at 12 months (86% vs. 61%; *p* < 0.01). However, primary patency was not different between the two groups at 24 months (52% vs. 55%; *p* = 0.48). No significant difference was found in the secondary patency rates at 12 (90% vs. 91%) and 24 (88% vs. 91%) months. At the 2-year follow-up point, the rate of percutaneous reintervention was similar; however, the s-AVFs had required more frequent surgical revision (36% vs. 17%; *p* = 0.01). Issues with wound healing and infection were also more frequent with s-AVFs (9% vs. 0.9%; *p* < 0.01).

## 5. Discussion

With an ageing population and longer survival from chronic diseases such as cancer, diabetes, and cardiovascular problems, end-stage renal disease (ESRD) has become a major burden for public health around the globe. Governmental policies and industry interests have focused on trying to better manage these problems and create innovative treatments that will improve patients’ lives and reduce costs while coping with the increasing number of patients requiring dialysis treatment.

Endovascular arteriovenous fistula creation is one such paradigm of disruptive technology, creating a major addition to vascular access options that has been little changed for more than fifty years. One question that frequently arises concerns the potential for the added cost of pAVF devices and whether this is justified in an environment where cost restriction is always a top priority when compared with alternative surgical procedures. Using Medicare data, Yang et al. attempted to address this issue in their publication by comparing 60 pAVF with 60 sAVF patients using a 1:1 propensity score matching baseline demographic and clinical characteristics [29]. They found that patients with a percutaneous fistula required fewer reinterventions post creation (0.59 vs. 3.43 per patient year; *p* < 0.05) with an overall savings of US$11,240 for the percutaneous group during the first year. While this amount would certainly suffice to cover the additional cost for the device during creation, it is worth mentioning that the number of reinterventions quoted for the surgical group appears excessive.

In a similar paper, data from the United States Renal Data System (USRDS) were abstracted to determine the rate of AVF interventions performed in the first year and associated costs for surgical AVFs (SAVF) created from 2011 to 2013 compared with data from the NEAT trial [30]. In the matched incident patients, the event rate was 0.74 per patient-year (PY) for endoAVF versus 7.22/PY for SAVF (*p* < 0.0001), with a difference in expenditures of $16,494 favouring endoAVFs. Similarly, in matched prevalent patients, the event rate was 0.46/PY for pAVF vs. 4.10/PY for sAVF (*p* < 0.0001), resulting in a cost difference of $13,389. While, again, the number of reinterventions for patients seems excessive, and data collection in any non-randomised study can always be prone to bias, those studies confirm the potential for cost savings in the post-creation period.

Our group’s experience after having created well over 300 hundred Ellipsys AVFs while also offering standard surgical procedures is that pAVF have several benefits. In addition to the immediate advantages of avoiding an incision and surgical dissection, less pain, and cosmetic advantages, radial artery inflow and multiple venous outflows make them hemodynamically much like the universal gold standard of a distal radio cephalic fistulae [31]. We previously reported [28] that pAVFs require more reinterventions during the first year to make them functional. Many of these reinterventions were related to the smaller size of the anastomosis in addition to the fact that dialysis personnel are familiar with the cannulation of single outflow high-pressure conduits. It has been globally observed and reported that once they become familiar with pAVFs, accessing the fistula becomes reliable. Much like a distal radio-cephalic fistula with multiple vein outflow, these modest flow and pressure pAVFs result in less intimal hypertrophy and high flow or pressure-related issues. Problems such as access-related hand ischemia, infections, aneurysm formation, bleeding episodes and high-flow congestive heart failure are very rare for pAVF and are a major consideration when comparing them with their surgical counterpart.

Even in the hands of an experienced pAVF provider, only about 50% of new patients will be candidates for a pAVF due to anatomical considerations, although this number may become higher with the physician’s experience. Many patients will need access created by traditional surgical techniques, and reliable pAVF creation procedures will not be available at many sites. Surgical access creation, revision and treatment of complications will continue to be critical for dialysis care.

As the number of percutaneous AVFs continues to increase worldwide, we plan to analyse cost differences and success rates for patients with pAVFs created at least three years previously and longer. Undoubtably, more data will be reported from multiple centres comparing surgical and percutaneous fistulas. Of course, a randomised trial would be needed to avoid inherent selection bias. Such a study would be difficult and may never be done. There are major differences between practice patterns and reimbursement systems in the United States and Europe; however, pAVF coding and electronic records will make comparisons easier. Clearly, individual practitioners will experience differing outcomes with these devices, particularly with initial use and the number of pAVFs created on an ongoing basis. Success with pAVFs will also be contingent on ultrasound expertise, close follow-up and interventional skills.

With experience, dialysis units and other health care practitioners involved in vascular access will become more familiar with this new type of AVF, and this will certainly help improve results. As always, technology will improve as well, directly impacting the devices and/or visualisation techniques, puncture methods or training. We remain optimistic that not only will patients’ lives and quality of care be improved with this innovative technology, but that cost savings for the healthcare system will also be achieved.

We all recall hesitancy and criticism of new techniques, such as endovascular grafts for aneurysms and laparoscopic and robotic surgery. Yet, they soon became the standard of care. Careful scientific evaluation of new medical techniques and devices is warranted; however, time and again, yesterday’s science fiction has proven to be tomorrow’s reality. Vascular access will not be an exception.

## Figures and Tables

**Figure 1 diagnostics-12-02447-f001:**
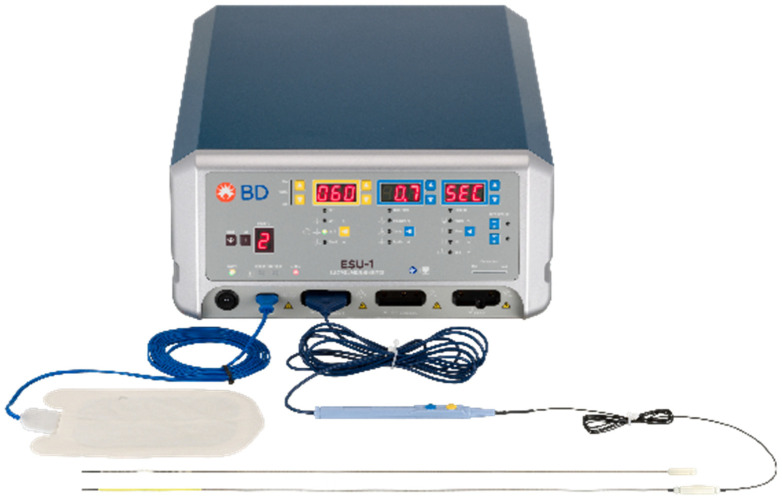
WavelinQ^TM^ EndoAVF Generator with Arterial and Venous Catheters.

**Figure 2 diagnostics-12-02447-f002:**
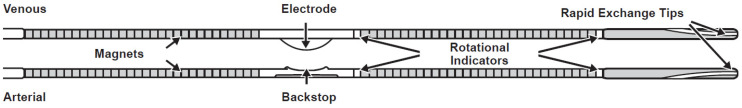
WavelinQ^TM^ Venous Catheter with Electrode and Arterial Catheter with Backstop.

**Figure 3 diagnostics-12-02447-f003:**
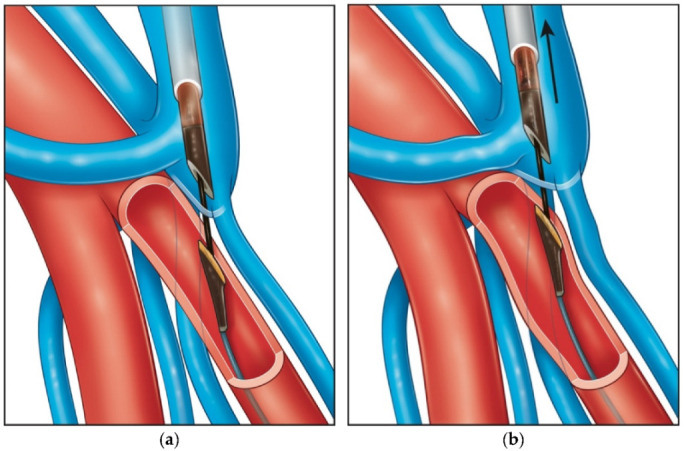
(**a**–**d**) stages of anastomosis creation with the Ellipsys device between the proximal radial artery and the perforating vein of the elbow—reproduced with permission from Mallios et al. J Vasc Surg [15].

**Figure 4 diagnostics-12-02447-f004:**
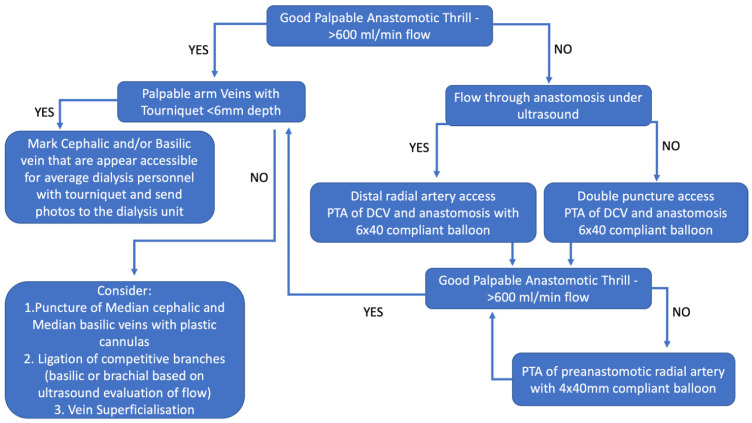
Algorithm for maintenance and assisted maturation for percutaneous fistula created with Ellipsys—reproduced with permission from Mallios et al. J Vasc Surg [20].

**Figure 5 diagnostics-12-02447-f005:**
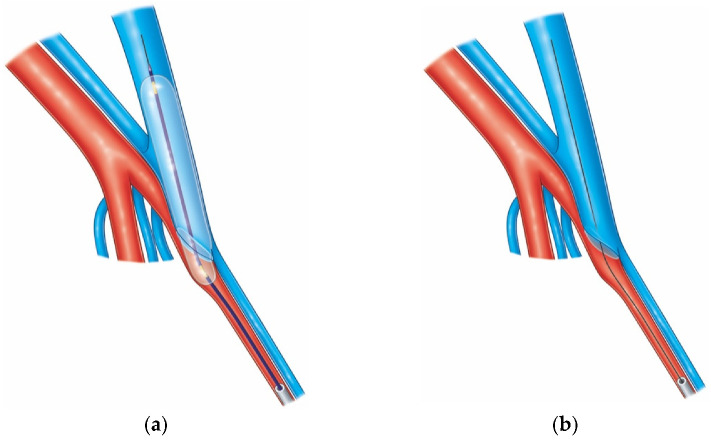
(**a**–**d**) Angioplasty of pAVF with radial access can allow dilatation of both outflow and inflow separately or simultaneously if needed with dual access reproduced with permission from Mallios et al. J Vasc Surg [20].

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
