# Peer review of "Endovascular Arteriovenous Fistula Creation—Review of Current Experience"

_diagnostics, 2022, doi:10.3390/diagnostics12102447_

Round 1

Reviewer 1 Report

Thank you for the opportunity to review your manuscript entitled "Endovascular arteriovenous fistula creation – review of current experience ".

The aim of the study is clear. The title is informative and relevant. The references are relevant, recent, and referenced correctly.

The manuscript is well written and a stimulus for the readership.

Author Response

Thank you for the time taken to review our paper 

Reviewer 2 Report

The authors present a review of endovascular fistula creation. They start by describing the technical process behind each one, with commentary based on their own experience, and finish with a brief summary of the available evidence. 

The section containing the evidence would be improved by some reorganization. Perhaps instead of talking about each trial, the section could be combined, with sections about topics such as patency, adverse events, and cost. This would provide a more cohesive, easier to read, flow to the paper. 

Additionally, technical tips and the author's opinions should likely appear under a header specifically denoting them, if present in the paper at all. 

Author Response

Thank you for the reviewing and recommendations for improvement, we are submitting the revises manuscript accordingly